# Evaluation and Characterization of Timber Residues of *Pinus* spp. as an Energy Resource for the Production of Solid Biofuels in an Indigenous Community in Mexico

**Mario Morales-Máximo** [1] , **Carlos A. García** [2] **, Luis Fernando Pintor-Ibarra** [1] **, José Juan Alvarado-Flores** [1] **, Borja Velázquez-Martí** [3] **and José Guadalupe Rutiaga-Quiñones** [1,*]

1    Facultad de Ingeniería en Tecnología de la Madera, Universidad Michoacana de San Nicolás de Hidalgo, Edif. D, Ciudad Universitaria, Morelia C.P. 58040, Michoacán, Mexico; mmoralesmaximo@gmail.com (M.M.-M.); luis.pintor@umich.mx (L.F.P.-I.); jjalvarado@umich.mx (J.J.A.-F.)
2    Escuela Nacional de Estudios Superiores, Unidad Morelia, Universidad Nacional Autónoma de México, Antigua Carretera a Pátzcuaro No. 8701, Col. Ex Hacienda de San José de la Huerta, Morelia C.P. 58190, Mexico; cgarcia@enesmorelia.unam.mx
3    Departamento de Ingeniería Rural y Agroalimentaria, Universidad Politécnica de València, Camino de Vera s/n, 46022 Valencia, Spain; borvemar@dmta.upv.es
*    Correspondence: rutiaga@umich.mx; Tel.: +52-(443)-322-3500 (ext. 3056)

**Abstract:** This study shows the energy potential of pine wood waste for the production of solid biofuels, and was carried out in an indigenous community in the state of Michoacán. One of the main economic activities of this community is the production of handcrafted furniture, which generates a large amount of wood waste. The most relevant results obtained in this research show that the community generates approximately 2268 kg of sawdust and 5418 kg of shavings per week, and the estimated energy potential per year for both sawdust is 1.94 PJ and for shaving is 4.65 PJ. Based on the particle size observed, the wood residue can be used to generate pellets or briquettes. Other average results in sawdust and (shavings) are the following: initial moisture content 15.3% (16.8%), apparent density 169.23 kg/m$^3$ (49.25 kg/m$^3$), ash 0.43% (0.42%), volatile material 84.9% (83.60%), fixed carbon 14.65% (15.96%), hemicelluloses 12.89% (10.68%), cellulose 52.68% (52.82%), lignin 26.73% (25.98%), extractives 7.69% (10.51%), calorific value 17.6 MJ/kg (17.9 MJ/kg). The major chemical elements in the ash were Al. K. Fe, Ca, P, Na, and Mg. Finally, the results obtained indicate that this biomass can be used to generate pellets or briquettes in this indigenous community.

**Keywords:** lignocellulosic residues; solid biofuels; bioenergy; briquettes; chemical analysis; ultimate analysis

## 1. Introduction

The use of renewable energy resources is essential. Over the years, various biomasses that can provide energy with less environmental impact have been discussed. The focus is on the catalytic conversion of wood biomass, with the aim of mitigating global warming, and reducing the emission of $CO_2$, through a potential catalyst, together with a sustainable concept for biorefineries based on lignocellulosic materials [1]. The carbon cycle and the energy transduced into it serve to feed the planet's living processes at an economical level of energy transfer using formed and neutral bonds between carbon molecules. These processes form the basis for the transfer of energy in the profitable production of fuel liquids from plant biomass; of particular importance are those aspects related to plant cell walls and their bioconversion [2] in different products, such as bioethanol [3]. Mexico is an underdeveloped country that has a low share of renewable energy sources in its energy matrix [4,5], since its energy system is based on fossil fuels; crude oil represents 59.8% of primary energy production, followed by natural gas with 24.10%, and coal with 3.64%, while renewable energies contribute 10.46% (solar 0.64%, wind 0.95% and biomass 5.70%)

of the total energy demand [6,7]. The increase in world population and economic growth have created a greater demand for fossil fuels, which present challenges as they are finite resources and have significant environmental impacts, such as global climate change [8,9]. A transition to renewable sources of energy can help address these issues, as well as aid in the improvement of social justice and economic equity in local areas [10]. Industries point to lignocellulosic biomass (wood, straw, cereal grains, etc.) as a raw material for energy production. Furthermore, the use of lignocellulosic biomass for energy production allows the use of small-scale cogeneration plants to obtain energy [11]. The urgent shift to clean and sustainable energy promotes a transition capable of meeting the energy needs of humanity and making a sustainable future possible [4,12], which is a central theme for science, politics, and public discourse throughout the world [13]. One important component of this shift is investment in technologies for the generation of renewable energy sources [14–16]. The use of these woody biomass raw materials for bioenergy is desirable, as they are derived from a renewable resource, are locally produced, and carbon neutral, which can help with energy security, reduce greenhouse gases, and create job opportunities that support rural development [17]. Studies on the characterization of biomass properties include proximal analysis, final analysis, and calorimetry [18,19], as they are of utmost importance to know the available useful energy and the transformation into densified solid biofuels [20,21]. The use of biomass as an alternative renewable energy source to reduce the amount of $CO_2$ emissions is relatively new, and accounts for about 10% of the total energy produced worldwide [22–25]. Biomass sources, such as forestry, depend on technical innovation and the broader social acceptance of these products, as they represent an emerging alternative renewable energy resource. The current methods of forest harvesting at a global level lie in sawing and wood processing activities, where large amounts of by-products are generated in the form of tree branches, tips, bark and sawdust. These by-products are commonly referred to as biomass, and are often used as combustibles [26]. In the case of Mexico, this biomass source comes from temperate forests that are mostly dominated by pine-oak species [27], which cover an area of 31.8 million hectares [28]. The total national forest production for 2018 was 8.3 million cubic meters of timber. The states of the Mexican republic with the highest production were: Durango (30.2%), Chihuahua (19.9%), Oaxaca (9.5%), Veracruz (6.1%), Michoacán (5.4%), and other states (28.9%). Forest production for the state of Michoacán in this same year by genus or types of wood had the following distribution: 85.1% (*Pinus* spp.), 6.2% (*Quercus* spp.), 5.3% (Mexican pine "Oyamel"), 3.4% (other hardwoods), 0.46% (common tropical woods), and 0.025% (other conifers). The total forest production of pine wood in the state of Michoacán was 85.1% and had the following distribution: 93.2% (squared timber), 6.6% (cellulosic material), 0.1% (posts, piling and fencing). The total forestry production of oak wood was 6.2%, with the following distribution: 69.6% (squared timber), 23.9% (cellulosic material), 4.0% (firewood), and 2.5% (charcoal) [29]. Forest residues for power generation have grown globally due to their potential as sources of clean, affordable, and renewable energy; however, knowledge about the availability of raw material, costs, and possible suppliers of biomass is scarce in Mexico [30]. This study identifies the potential for timber waste in the indigenous community of Pichátaro, located in the state of Michoacán, Mexico. The forest of this community is made up of species of pine and oak. The predominant pine species are *Pinus pseudostrobus*, *P. montezumae*, and *P. leiophylla*. In the case of oak woods, the most representative species are *Quercus rugosa* and *Quercus laurina* [31]. No current data were available on the trees density per hectare or the forest area of the indigenous community, but for the year 2014 the forest area was approximately 8484 hectares [32]. In this community, the main economic activity is the transformation of wood into finished products, mainly furniture. This activity involves approximately 200 artisan workshops, in which rustic furniture and other artisan articles are produced (for example: wardrobes, chairs, dining rooms, board games, and animal figures, among others). It should be noted that this production is usually manual, and uses manual tools such as handsaws, scroll saws, circular saws, edgers, roller sanders, framing chisels, jackplanes, and others. This

artisanal process produces sawdust and shavings as its main waste product. It is important to know the amount of waste that is generated in this indigenous community, and to determine its physical, chemical, and energy properties, because this timber waste can be different from others, either because of the origin of the raw material or through the type of processing. It is known that wood processing into finished products generates a large amount of wood waste, which is underused or not used at all, and produces atmospheric pollution. The amount of this residue is not quantified, therefore the potential or scope of the development of solid biofuels and its application and benefit to rural communities is unknown [33,34]. For this reason, this article explores the possibilities of using wood biomass of *Pinus* spp., since by determining the physical, chemical, and energetic properties of the waste generated in the studied community, it may aid the development and small scale production of densified materials, satisfy the demand for local thermal energy, and thereby contribute to the generation of biofuel solids derived from waste, thus generating a lower environmental impact for the planet. Until now, there has been no large-scale production or use of wood-based biomasses in Mexico and there are no detailed studies on their technical energy potential, logistics costs, or specific uses [35,36]. In addition to the energy potential of the wood waste, it is important to understand that the biomass residues have the potential to meet the energy demands within the same communities. The rural communities involved can benefit from the increased value of the timber residue, which will create a new industry and cause the living conditions of the communities involved to improve.

## 2. Materials and Methods

### 2.1. Experimental Section

#### 2.1.1. Community Diagnosis

The study community was San Francisco Pichátaro (latitude 19.55°, longitude 101.8°) located in the state of Michoacán, Mexico [37]. In this indigenous community there are approximately 200 artisan workshops that transform wood into typical furniture, of which a sample of 70 artisans was randomly chosen, representing 65% of the total. Chosen artisans have a common characteristic: their workshops have the basic tools to work, in addition to the availability to provide information for this research. Through the application of surveys and home visits, data were obtained on the type of wood used to make furniture and the amount of wood waste they generate. Subsequently, of the 70 workshops surveyed, only 10 workshops were chosen, and as the community is divided into 7 neighborhoods, 1 workshop was chosen for each neighborhood, the other three workshops were chosen at random. From this last sampling, 5 sawdust and 5 shavings samples were obtained for analysis.

#### 2.1.2. Moisture Content

The initial moisture content of the sawdust and shavings was determined in triplicate by the dehydration method according to the UNE-EN 14774-1 [38]. Subsequently, the collected samples were allowed to dry in the open air for 4 weeks, then, a representative portion of each sample of the wood residues was ground using Wiley equipment to obtain 40-mesh woodmeal, which was used for the analyses described below.

#### 2.1.3. Granulometry

The particle size distribution of the sawdust and shavings samples dried in the open air was determined following the UNE-EN 15149-1 standard [39]; using a vibrating sieve (RoTap RX-29), the sieving process it was 3 min.

#### 2.1.4. Bulk Density

The bulk density of the sawdust and shavings samples was determined by the UNE-EN 15103 [40] standard, using an AS 310.R2 PLUS analytical balance.

### 2.1.5. Proximate Analysis

The ash content of the timber residues was determined according to the EN 14775 standard [41], and the volatile matter content according to the ASTM E872-82 standard [42]. For this case, absolutely dry 40-mesh woodmeal was used. Fixed carbon was calculated by difference, subtracting the ash content and volatiles by 100% [18].

### 2.1.6. Ultimate Analysis

The content of carbon, hydrogen, nitrogen, and sulfur was measured in an elemental analyzer (Model 4010; Costech International S.p.A., Milan, Italy) following the UNE-CEN/TS 15104 EX standard [43]. For this case, absolutely dry 40-mesh (425 μm) biomass was used. The oxygen content was calculated by difference [44], and the analysis was performed only once.

### 2.1.7. Basic Chemical Analysis

To determine the chemical composition of the timber residues (sawdust and shavings), a fiber analysis was carried out based on the gravimetric method Van Soest using α-amylase in an ANAKOM-200 equipment [45]. For this purpose, absolutely dry 40-mesh (425 μm) woodmeal [46] was used.

### 2.1.8. Ash Microanalysis

A Varian inductively coupled plasma-optical emission spectrophotometer (ICP-AES) (Model 730-ES; Varian Inc. (Agilent), Mulgrave, Australia) was used to carry out the ash microanalysis from the wood residue samples [47]. The ash microanalysis searched for the presence of 29 chemical elements and was only conducted once.

### 2.1.9. Calorific Value

The calorific value of the sawdust and shavings samples was determined using a semiautomatic calorimeter (LECO AC600, St. Joseph, MI, USA) according to EN-14918 [48]. For this purpose, absolutely dry 100-mesh (425 μm) woodmeal was used and the analysis was carried out in duplicate.

### 2.1.10. Community Energy Potential

The energy potential of biomass is obtained from the relationship that exists between the mass of dry waste (*Mrs*) and the energy of the waste per unit mass (*E*), also known as calorific value (CV). Equation (1) expresses the relationship between the variables and proposes an approximate mathematical model [49].

$$PE = (Mrs) \times (E) \tag{1}$$

where:

*PE*: Energy potential [TJ/year]
*Mrs*: Mass of dry residue [t/year]
*E*: Energy of the residue per unit mass [TJ/t]
CV: Calorific value (MJ/kg)

### 2.1.11. Statistical Analysis

In order to compare the data obtained in analyses that were conducted more than once, an analysis of variance was performed at 95% statistical confidence and the mean values were compared using the multiple range test with the method of least significant difference (LSD) [50]. The data obtained was processed using Statgraphics Centurion 19.2.01. In all cases, the average value and standard deviation are reported.

2.1.12. Multi-Criteria Analysis

Multicriteria analysis is a tool to support decision-making during a certain process that allows the integration of different criteria according to the opinion of actors in a single analysis framework to provide a comprehensive vision [51]. This study utilized multi-criteria analysis to take into account sustainability indicators, considering the methodology of multicriteria analysis and sustainability evaluation with the help of the MULTIBERSO program [52].

## 3. Results

### 3.1. Community Diagnosis

The data obtained in the diagnosis show that the species most used to obtain wood used in the manufacture of furniture correspond to the *Pinus* genus with 90%, followed by oak wood (*Quercus* spp.) with 10%. In addition, the diagnosis indicates that 2268 kg of sawdust ($\pm$15.2) and 5418 kg of shavings ($\pm$17.8) of *Pinus* spp. are generated per week on average for each workshop, with sawdust representing 30.4% and shavings 69.6% of total waste. Taking into account the 200 artisan workshops, it is estimated that the amount of timber waste generated per week is 158,760 kg of sawdust and 379,260 kg of shavings.

### 3.2. Moisture Content

The result of the analysis of variance indicates that there are significant statistical differences ($p$ = 0.0033), as observed in the graph of means (Figure 1). The initial moisture found in the sawdust samples ranged from 14.21% ($\pm$0.16) to 21.9% ($\pm$0.29) with an average value of 15.30% ($\pm$3.83).

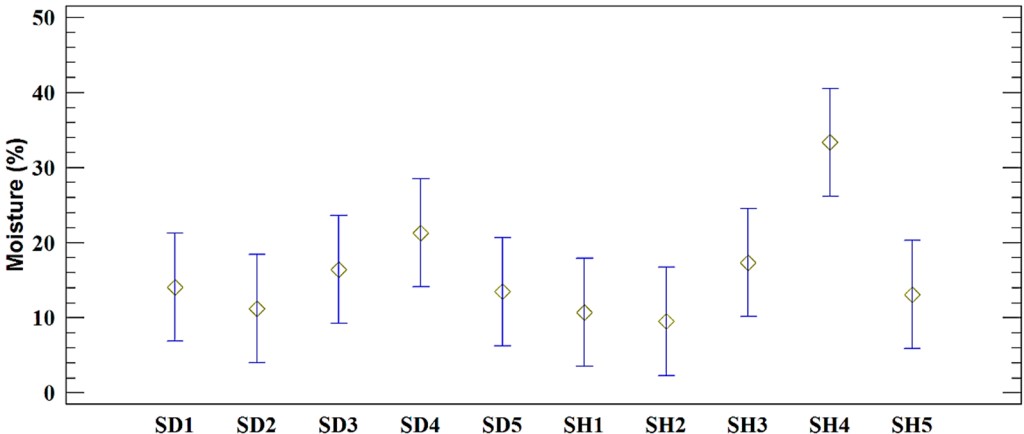

**Figure 1.** Initial moisture content of the sawdust (SD) and shavings (SH) samples (%).

### 3.3. Granulometry

The particle size found in the sawdust samples (SD) can be seen in Figure 2, and Figure 3 shows the particle size found in the shavings samples (SH). These values correspond to the average of the five samples taken for each type of material.

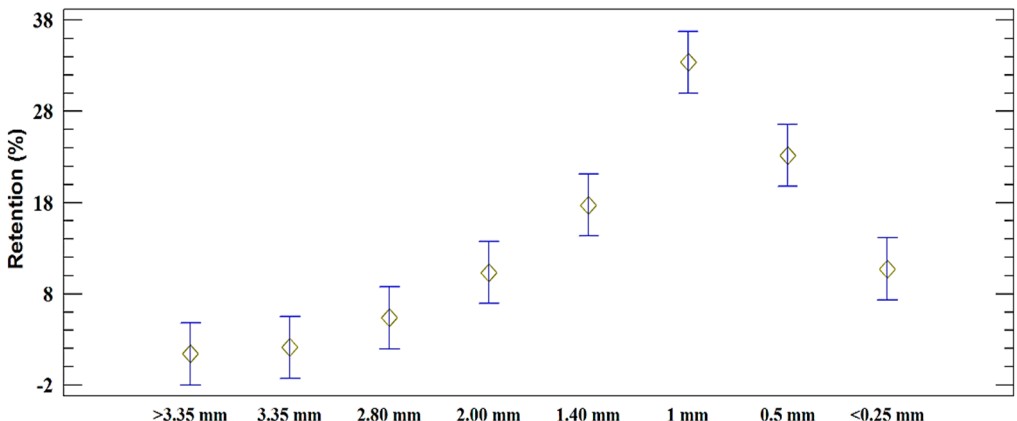

**Figure 2.** Granulometric distribution of the sawdust samples (%).

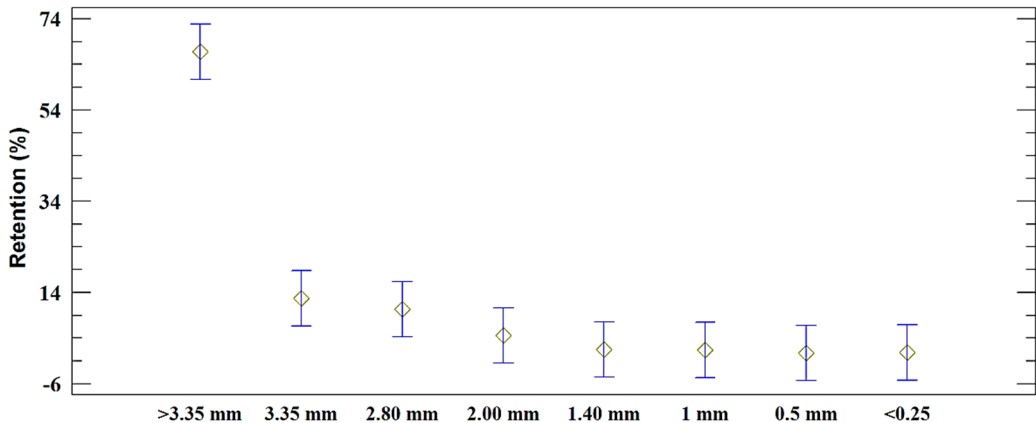

**Figure 3.** Granulometric distribution of the shavings samples (%).

### 3.4. Bulk Density

The result of variance analysis for density indicates that there are statistically significant differences ($p = 0.0000$). The result for the sawdust samples ranged from 143.89 ($\pm 5.84$) kg/m$^3$ to 196.91 ($\pm 13.90$) kg/m$^3$ (Figure 4) with an average value of 169.23 ($\pm 23.64$) kg/m$^3$. The bulk density for the shavings samples ranged from 40.78 kg/m$^3$ ($\pm 3.52$) to 57.88 kg/m$^3$ ($\pm 3.10$) with an average value of 49.25 kg/m$^3$ ($\pm 8.16$).

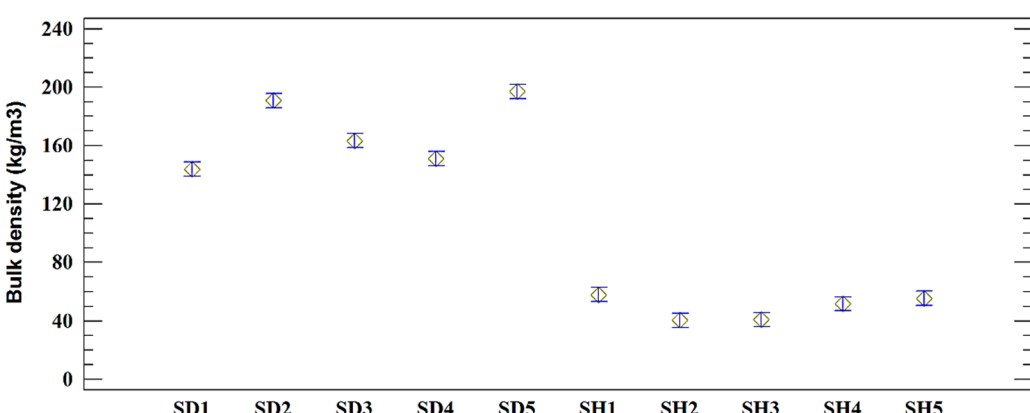

**Figure 4.** Bulk density of the sawdust (SD) and shavings (SH) samples (kg/m$^3$).

### 3.5. Proximate Analysis

The result of variance analysis indicates that there are no statistically significant differences ($p = 0.7820$). The ash content in the sawdust samples ranges from 0.30% ($\pm 0.33$)

to 0.64% (±0.39) with an average value of 0.43% (±0.14). The ash content in the shavings samples ranges between 0.31% (±0.36) and 0.59% (±0.36), with an average value of 0.42% (±0.11) (Figure 5).

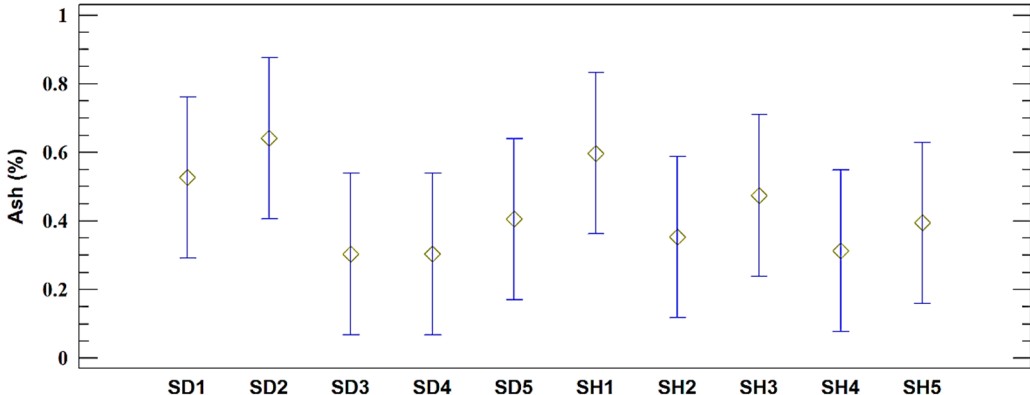

**Figure 5.** Ash content in the sawdust (SD) and shavings (SH) samples (%).

The ash content for the shavings samples ranged from 66.91% (±1.02) to 90.88% (±1.05), with an average value of 83.60% (±9.80) (Figure 6).

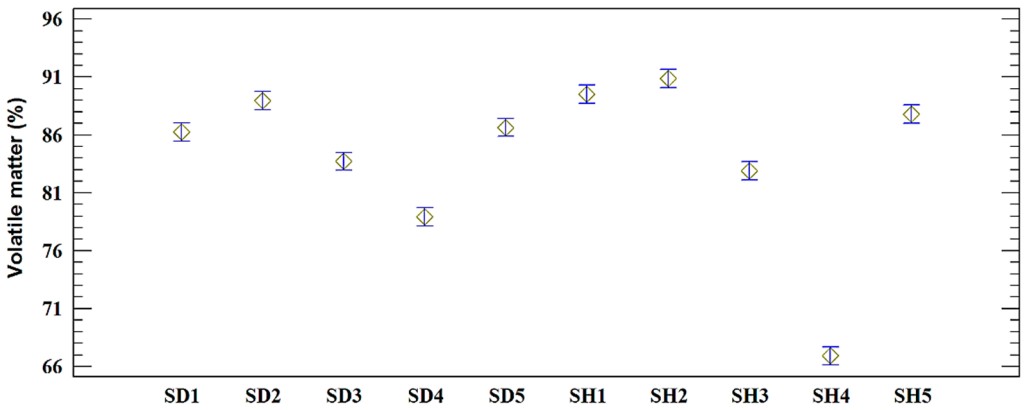

**Figure 6.** Volatile matter for the sawdust (SD) and shavings (SH) samples (%).

In relation to fixed carbon (Figure 7), the analysis of variance indicates that there are statistically significant differences ($p = 0.0000$).

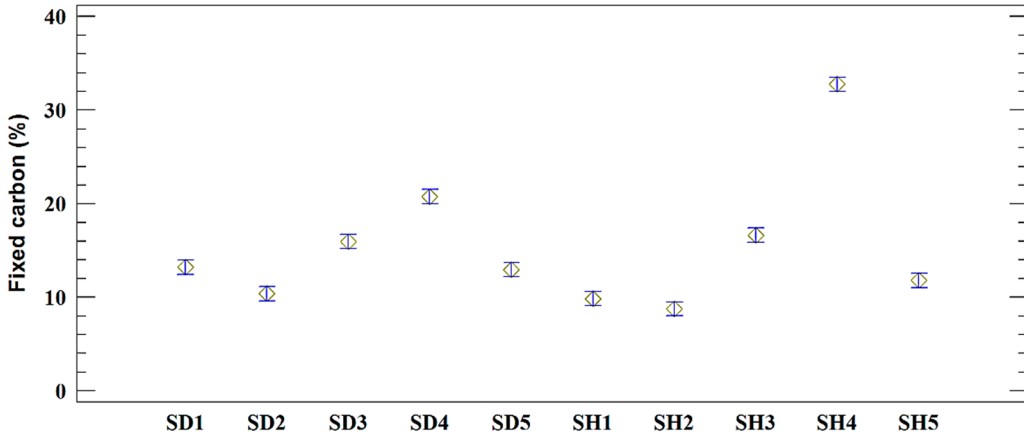

**Figure 7.** Fixed carbon for the sawdust (SD) and shavings (SH) samples (%).

### 3.6. Ultimate Analysis

The results of the elemental analysis of the sawdust (SD) and shavings (SH) samples are shown in Table 1.

**Table 1.** Ultimate analysis of sawdust (SD) and shavings (SH) samples (%).

| Samples | C | H | O | N | S |
|---------|---|---|---|---|---|
| SD1 | 47.73 | 6.09 | 45.45 | 0.71 | <0.01 |
| SD2 | 47.98 | 6.13 | 45.49 | 0.37 | <0.01 |
| SD3 | 48.79 | 5.98 | 44.79 | 0.41 | <0.01 |
| SD4 | 48.12 | 6.09 | 45.37 | 0.40 | <0.01 |
| SD5 | 48.29 | 6.00 | 45.26 | 0.43 | <0.01 |
| Average | 48.18 (±0.39) | 6.06 (±0.06) | 45.27 (±0.28) | 0.46 (±0.13) | <0.01 |
| SH1 | 47.78 | 5.99 | 45.60 | 0.60 | <0.01 |
| SH2 | 48.31 | 6.16 | 45.12 | 0.38 | <0.01 |
| SH3 | 47.73 | 6.04 | 45.81 | 0.38 | <0.01 |
| SH4 | 47.96 | 6.05 | 45.56 | 0.41 | <0.01 |
| SH5 | 48.96 | 6.09 | 44.50 | 0.43 | <0.01 |
| Average | 48.15 (±0.50) | 6.06 (±0.06) | 45.32 (±0.52) | 0.44 (±0.09) | <0.01 |

### 3.7. Basic Chemical Analysis

Table 2 shows the result of the chemical analysis of the sawdust (SD) and shavings (SH) wood residues. Regarding the sawdust samples, the average result was: hemicellulose (12.89%, ±2.64), cellulose (52.68%, ±2.82), lignin (26.73%, ±0.89), and extractives (7.69%, ±1.34). For the case of the shavings samples, the average result was: hemicellulose (10.68%, ±2.05), cellulose (52.82%, ±1.73), lignin (25.98%, ±0.99), and extractives (10.51%, ±1.16).

**Table 2.** Basic chemical composition of sawdust (SD) and shavings (SH) samples (%).

| Samples | Hemicellulose | Cellulose | Lignin | Extractives |
|---------|---------------|-----------|--------|-------------|
| SD1 | 17.34 | 48.16 | 25.60 | 8.9 |
| SD2 | 12.41 | 54.87 | 26.88 | 5.84 |
| SD3 | 11.39 | 54.54 | 26.94 | 7.13 |
| SD4 | 12.83 | 51.69 | 28.01 | 7.47 |
| SD5 | 10.49 | 54.17 | 26.23 | 9.11 |
| Average | 12.89 (±2.64) | 52.68 (±2.82) | 26.73 (±0.89) | 7.69 (±1.34) |
| SH1 | 10.34 | 51.89 | 26.01 | 11.76 |
| SH2 | 10.51 | 54.79 | 25.95 | 8.75 |
| SH3 | 9.12 | 53.86 | 25.83 | 11.19 |
| SH4 | 9.24 | 53.24 | 27.46 | 10.06 |
| SH5 | 14.19 | 50.35 | 24.66 | 10.8 |

### 3.8. Ash Microanalysis

Table 3 shows the result of the microanalysis of the ash from the sawdust (SD) and shavings (SH) samples.

**Table 3.** Ash microanalysis of sawdust (SD) and shavings (SH) samples (ppm).

| Element | SD1 | SD2 | SD3 | SD4 | SD5 | SH1 | SH2 | SH3 | SH4 | SH5 |
|---|---|---|---|---|---|---|---|---|---|---|
| Ag | ND | ND | ND | ND | ND | ND | ND | ND | ND | ND |
| Al | 15,883.66 | 4770.21 | 19,198.40 | 16,808.97 | 16,211.77 | 10,390.06 | 15,397.57 | 14,295.14 | 12,238.74 | 22,787.87 |
| As | ND | ND | ND | ND | ND | ND | ND | ND | ND | ND |
| B | 22.23 | 148.20 | 69.71 | 115.31 | 48.15 | 21.26 | 85.56 | 15.76 | 94.17 | 88.92 |
| Ba | 115.58 | 292.05 | 183.96 | 162.40 | 114.96 | 131.66 | 203.33 | 92.94 | 158.19 | 200.81 |
| Be | ND | ND | ND | ND | ND | ND | ND | ND | ND | ND |
| Ca | 3946.68 | 24,350.40 | 8771.71 | 14,685.20 | 8720.03 | 2671.08 | 16,015.29 | 7234.63 | 14,357.56 | 15,801.00 |
| Cd | 1.57 | 1.32 | 2.25 | 2.34 | 1.75 | 2.01 | 2.54 | 1.10 | 1.82 | 2.93 |
| Co | ND | ND | ND | ND | ND | ND | ND | ND | ND | ND |
| Cr | 11.54 | ND | 9.22 | 10.66 | 8.20 | 15.19 | 7.27 | 3.57 | 6.91 | 16.98 |
| Cu | 70.21 | 230.36 | 156.34 | 223.88 | 126.50 | 117.05 | 420.09 | 58.18 | 236.96 | 415.85 |
| Fe | 5114.25 | 2110.25 | 7834.38 | 8357.62 | 7647.26 | 4696.31 | 7862.51 | 6923.31 | 6633.44 | 11,632.98 |
| K | 12,776.78 | 96,001.30 | 29,193.39 | 50,875.25 | 30,390.46 | 8301.61 | 56,352.52 | 27,200.89 | 48,529.56 | 58,896.15 |
| Li | 9.35 | 105.45 | 31.74 | 38.03 | 24.42 | 32.26 | 29.15 | 31.66 | 33.24 | 6.32 |
| Mg | 1312.76 | 8714.68 | 3034.71 | 5224.66 | 3127.17 | 819.25 | 5332.31 | 2612.19 | 4816.56 | 5531.51 |
| Mn | 287.62 | 1031.42 | 574.89 | 928.14 | 402.66 | 318.79 | 730.33 | 281.38 | 646.38 | 694.20 |
| Mo | ND | ND | ND | ND | ND | ND | ND | ND | ND | ND |
| Na | 2368.42 | 3636.02 | 2200.23 | 2121.57 | 1723.90 | 1513.92 | 2734.95 | 2220.94 | 2229.47 | 2313.20 |
| Ni | 22.60 | 19.82 | 64.48 | 59.41 | 34.14 | 40.22 | 40.92 | 15.78 | 23.19 | 30.74 |
| P | 5892.63 | 8477.57 | 5159.20 | 10,391.33 | 4566.17 | 2625.44 | 6984.77 | 2055.79 | 6442.35 | 6885.04 |
| Pb | ND | ND | ND | ND | ND | ND | ND | ND | ND | ND |
| Sb | ND | ND | ND | ND | ND | ND | ND | ND | ND | ND |
| Se | ND | ND | ND | ND | ND | ND | ND | ND | ND | ND |
| Si | 287.89 | 3864.95 | 234.59 | 6050.14 | 338.16 | 174.93 | 7548.83 | 18.01 | 50.05 | 1501.70 |
| Sn | 2.76 | 16.96 | 2.26 | 20.47 | ND | 8.12 | 3.97 | ND | ND | 2.57 |
| Sr | 102.81 | 452.39 | 240.41 | 350.54 | 178.83 | 104.16 | 280.28 | 97.79 | 346.09 | 326.90 |
| Tl | ND | ND | ND | ND | ND | ND | ND | ND | ND | ND |
| V | 28.07 | 6.39 | 29.08 | 28.22 | 27.10 | 35.02 | 26.73 | 21.97 | 18.87 | 37.25 |
| Zn | 245.44 | 2557.69 | 535.13 | 614.66 | 340.92 | 241.61 | 1058.91 | 181.06 | 571.45 | 696.06 |

ND = not detected.

### 3.9. Calorific Value

The average calorific value for the sawdust samples was 17.6 MJ/kg (±0.15), and for the shavings samples 17.9 MJ/kg (±0.15).

### 3.10. Community Energy Potential

According to the diagnosis of the availability of the timber residue in the indigenous community, approximately 108.86 tons per year of sawdust and 260.06 tons per year of shavings are generated. Thus, the energy potential for this indigenous community is as follows: for sawdust it is 1.94 PJ/yr, and for shavings it is 4.65 PJ/yr.

### 3.11. Multi-Criteria Analysis

This analysis considered energy, physical-proximal, and chemical composition parameters, which lead to quantifiable indicators, which can be seen in Table 4.

**Table 4.** Parameters and indicators used in the multi-criteria analysis.

| Parameter | Indicator |
|---|---|
| Energetic | Calorific value (MJ/Kg) |
| | Moisture (%) |
| Physical-proximal | Ash (%) |
| | Volatile matter (%) |
| | Fixed carbon (%) |
| | Lignin (%) |
| Chemical composition | Cellulose (%) |
| | Hemicellulose (%) |
| | Extractives (%) |

The weighting with maximum and minimum values that define the best and worst scenario, the maximum value data are obtained from the scientific literature, as shown in Table 5.

**Table 5.** Indicator values.

| Indicator | Maximum Value | Minimum Value |
|---|---|---|
| Calorific value (MJ/Kg) | 20.92 [53] | 0 |
| Moisture (%) | 56 [54] | 0 |
| Ash (%) | 18.20 [53] | 0 |
| Volatile matter (%) | 86.32 [55] | 0 |
| Fixed carbon (%) | 66.16 [56] | 0 |
| Lignin (%) | 43.37 [57] | 0 |
| Cellulose (%) | 53.3 [58] | 0 |
| Hemicellulose (%) | 87.11 [59] | 0 |
| Extractives (%) | 56.1 [60] | 0 |

The multi-criteria methodology is not a *per se* tool, it must always be comparative. In this study, the following timber residues were analyzed for comparison: (1) pine residue (*Pinus* spp.) and (2) oak residue (*Quercus* spp.). The real evaluation of the indicators for the two case studies are shown in Table 6. These values (Table 6) are normalized with the values in Table 5, to establish a scale from 0 to 10. Zero represents the worst possible scenario, while ten the best. The normalized values are shown in Table 7.

**Table 6.** Evaluation of the indicators.

| Indicator | *Pinus* spp. Waste | *Quercus* spp. Waste |
|---|---|---|
| Calorific value (MJ/Kg) | 18.0 | 19.5 [26] |
| Moisture (%) | 16.82 | 25 [26] |
| Ash (%) | 0.64 | 0.95 [26] |
| Volatile matter (%) | 78.92 | 87.33 [26] |
| Fixed carbon (%) | 8.76 | 8.88 [26] |
| Lignin (%) | 26.7 | 24.5 [61] |
| Cellulose (%) | 52.6 | 38.4 [61] |
| Hemicellulose (%) | 13.14 | 24 [61] |
| Extractives (%) | 19.8 | 6.94 [62] |

**Table 7.** Normalized values for the analysis.

| Indicator | *Pinus* spp. Waste | *Quercus* spp. Waste |
|---|---|---|
| Calorific value (MJ/Kg) | 8.60 | 9.32 |
| Moisture (%) | 3.00 | 4.46 |
| Ash (%) | 0.18 | 0.27 |
| Volatile matter (%) | 9.14 | 10.11 |
| Fixed carbon (%) | 1.32 | 1.34 |
| Lignin (%) | 6.15 | 5.64 |
| Cellulosa (%) | 9.86 | 7.20 |
| Hemicellulose (%) | 1.50 | 2.75 |
| Extractives (%) | 3.52 | 1.2 |

The multi-criteria analysis and sustainability indicators, which show the results graphically, are shown in Figure 8.

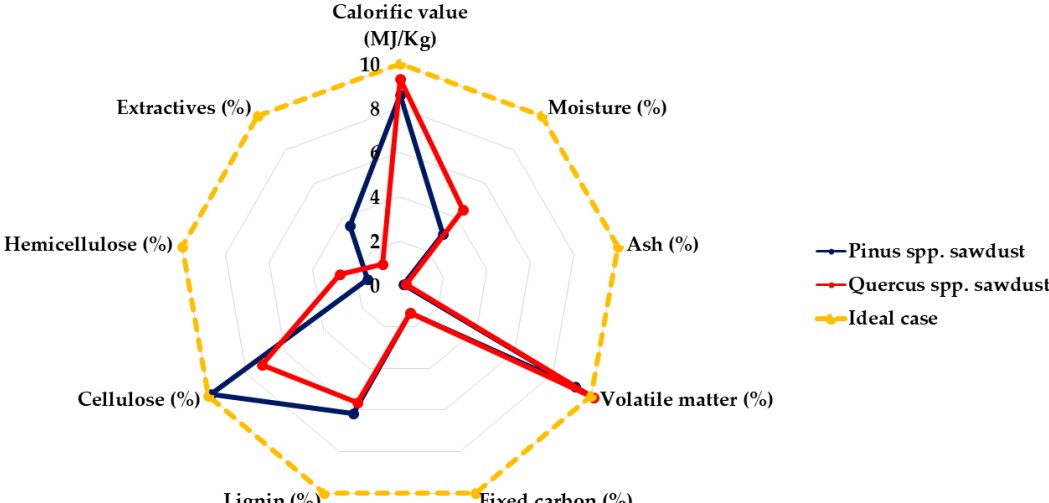

**Figure 8.** Formulation of indicators with different parameters.

The different indicators that intervened in the respective analysis are shown in Figure 8; this allows comparison of the potential of the *Pinus* spp. timber residue with that of the *Quercus* spp. timber residue. In a graphic way, strengths and weaknesses can be appreciated in different aspects, from an energetic, physical-proximal, and chemical composition approach.

## 4. Discussion

### 4.1. Moisture Content

Data reported in the scientific literature indicate an average moisture content of the sawdust ranging from 15% to 37% [63,64]. For the shavings samples, the moisture ranged from 9.54% (±0.78) to 33.55% (±39.26) with an average value of 16.82% (±9.70). These results obtained are within the range of 10% to 60% of the average moisture content reported for biomass [54,65,66], which indicates that an artificial drying process would not be necessary to use it in the elaboration of densified biofuels.

The moisture content reached by the samples in the open air after 4 weeks was 15.30% (±3.83) for sawdust, which is in the range reported for dry sawdust in the open air (5% to 19%) [67]. For the shavings samples, the result was 16.82% (±3.23), and previous reports indicate at least 8.4% [68] and 9.2% [69], indicating that the final moisture content in the samples studied here is ideal for densified biofuel manufacturing, without the need for additional drying.

### 4.2. Granulometry

The granulometric analysis for the sawdust indicates that approximately 70% of particles were greater than 1.00 mm, therefore, practically all of the sawdust could be used to make pellets, since the optimal size of biomass for pellets ranges from 1 mm to 5 mm [70]. In the case of shavings, approximately 97% of the particles were recorded to be larger than 1.00 mm, while other studies report that the proportion of pine sawdust for briquettes varies from 30% to 80% [71], therefore, the shavings could be used for the development of briquettes, although this will depend on the briquetting process. Another option for the production of biofuels densified with these two lignocellulosic materials could be to prepare mixtures of both materials and seek the best properties. There is evidence that using mixtures of different particle sizes results in desirable properties, most notably in pellets [72,73].

### 4.3. Bulk Density

The sawdust bulk density coincides with mean values between $160 \, \text{kg/m}^3$ to $235 \, \text{kg/m}^3$ previous reports [53,74]. For the case of shavings, values from $160 \, \text{kg/m}^3$ to $330 \, \text{kg/m}^3$ are reported [65]. The bulk density for the shavings samples in this investigation is within acceptable parameters. The densities reported in pines are between $420 \, \text{kg/m}^3$ and $670 \, \text{kg/m}^3$ [23].

### 4.4. Proximate Analysis

Our proximate analysis results generally coincide with the data reported for pine woods: 0.28% to 1.25% [54], 0.38% to 1.78% [58], 0.22% to 1.92% [75], 1.07% [76], and 0.27% to 0.95% [77]. The amount of ash in biomass plays an important role when designing thermal installations to burn densified biofuels, and also influences the calorific value [78]. According to [79] our study material could be used to produce class A1 pellets, whose requirement is an ash content of less than 0.7%.

Regarding the content of volatile matter, the result of the analysis of variance indicates that there are significant statistical differences ($p = 0.0000$). The volatile matter content for the sawdust samples was found to range from 78.92% ($\pm 1.03$) to 86.64% ($\pm 1.04$), with a mean value of 84.90% ($\pm 3.82$), which is similar to those reported for pine sawdust (78.9% to 89.8%) [77]. The results found for volatile matter generally coincide with previously reported values for different timber residues (80.84 to 81.27%) [80], 78.60% [81], and 91.26% to 95.01% for shavings samples, and 65.3% to 90.29% for sawdust samples [58], 78.60% [81].

The fixed carbon values for sawdust ranged from 10.38% ($\pm 0.67$) to 20.77%, ($\pm 0.72$) with an average of 14.65% ($\pm 3.94$), which coincide with data reported for pine sawdust samples (9.6 to 20.4%) [77]. The results for the shavings samples ranged from 8.71% ($\pm 0.67$) to 32.76% ($\pm 0.82$) with an average of 15.96% ($\pm 9.86$). Other investigations report different values for timber residues: 48.80% to 50.30% [82], 12.20% [83], 16.76% [84], 15.96% [85], and from 4.49% to 34.35% [58]. In samples of agricultural residues and pine sawdust they report 15.54% of fine carbon [86].

### 4.5. Ultimate Analysis

These results are in general agreement with the values reported for different types of biomass [64]. The percentage of nitrogen and sulfur in the wood is usually low, as the results show. Carbon and oxygen are the main components of solid biofuels and are those that participate in the exothermic reaction during combustion, generating $CO_2$ and $H_2O$ [70]. For the biomass samples of sawdust, the carbon content ranged between 47.73% and 48.29%, while the samples of shavings ranged between 47.73% and 48.96%. The oxygen content for the sawdust was between 44.79% and 45.49%, while for the shavings it ranged between 44.50% to 45.81%. Finally, the hydrogen content in sawdust ranged from 5.98% to 6.13%, and for shavings it was between 5.99% and 6.16%. The results obtained are consistent with those previously reported by recent studies [77,87,88].

### 4.6. Basic Chemical Analysis

Our chemical analysis results are similar to those previously reported for pine sawdust [58]. The chemical analysis results for shavings samples are in agreement with different studies for pine wood residues [89–95]. Lignin is the fundamental structural element for the generation of energy. In this investigation, we obtained results similar to the works reported for the species of *Pinus* genus, which is situated at 25.9% to 26.7% of lignin [96,97].

### 4.7. Ash Microanalysis

The ash analysis is useful for the characterization of biomass [78]. Twenty chemical elements were detected; the majority were Al, K, Fe, Ca, P, Na, and Mg. This result reveals the majority presence of the most common chemical elements present in wood (potassium, calcium, and magnesium), the main inorganic substances in wood [98], found up to 80% in ash [99]. The concentration of K in the wood samples is relatively high, and could generate problems in the combustion equipment [100,101], but this ash could also be used as fertilizer [102].

### 4.8. Calorific Value

The results of calorific value are acceptable for pine wood waste used for briquettes processing [103]. Other studies report values of 17.0 MJ/kg to 18.3 MJ kg [104], and of 16.91 MJ/kg for *Pinus* spp. [105].

### 4.9. Community Energy Potential

Case studies in Mexico report that the energy potential from the use of biomass for today and the two decades to come is on average 2228 PJ/yr [33]. Another study carried out in Mexico reports the potential of bioenergy and costs of the use for energy of woody forest biomass on a regional scale: an available theoretical energy of 45.96 PJ for the year 2013 [106]. Finally, for a regional case, focusing on three species with the highest utilization rates (*Pinus*, *Quercus*, and *Abies*), a forecast analysis was carried out for the year 2023, resulting in the calculation of potentially 60.22 PJ. This result meets the goals set by the National Forestry Commission of Mexico on hectares under sustainable use [29,107]. Mexico is not the only country in Latin America that estimates the energy potential of its biomass to supply its primary energy needs by generating energy through woody biomass from forest residues. Countries such as Colombia have an obvious interest in biomass; which in 2009 contributed 3.4 PJ in electricity generation, 15.7 PJ of energy supply in the transportation sector, and 193.5 PJ toward the total primary energy supply [108].

### 4.10. Multi-Criteria Analysis

The calorific value indicator shows that similarity by not presenting a significant difference. Regarding the moisture content, it can be seen that the *Pinus* spp. residue is lower, compared to the *Quercus* spp. residue, therefore no prior drying is required to stabilize the samples and produce densified solid biofuels. The ash content of the two residues previously mentioned is low; which makes them ideal for combustion in different domestic scenarios, in particular the residue of *Pinus* spp. The content of volatile matter is similar in both residues, with a slight difference, which is that the *Quercus* spp. residue contains a higher content of volatile material that can be harmful to the environment. Regarding fixed carbon, there is no significant difference, as the two residues have very similar values on average: 8.82%. For the lignin content, the indicators involved report similar values, although a higher percentage of lignin is found in the *Pinus* spp. residue. This higher lignin content directly influences the calorific value, which may be higher in this type of residue compared to *Quercus* spp. In the case of cellulose, the *Pinus* spp. residue contains a larger amount compared to *Quercus* spp. By contrast, the hemicellulose content is found in a higher percentage in the *Quercus* spp. residue. In summary, *Pinus* spp. timber waste represents an abundant and competitive raw material compared to *Quercus*

spp. timber waste in energy, physical-proximal, and chemical composition aspects, and is seen as a raw material for the generation of bioenergy with indicators of sustainability. For this indicator, the *Pinus* spp. timber residue is the ideal case.

## 5. Conclusions

The most relevant results obtained in this research show that the indigenous community (Pichátaro, Michoacán, Mexico) generates approximately 2268 kg of sawdust and 5418 kg of shavings per week, and the estimated energy potential per year for both sawdust is 1.94 PJ and for shaving is 4.65 PJ. This wood waste can be a valuable energy source that can partially or totally meet the energy demand of this community. This biomass, which is now underutilized, can be used, for example, to generate thermal energy. Of course, these lignocellulosic residues could also be used in the future for the production of densified solid biofuels (pellets or briquettes) for the indigenous community, or for other rural communities, with less emission of pollutants into the environment.

**Author Contributions:** M.M.-M.; conceptualization; M.M.-M., C.A.G. and J.G.R.-Q.; conceived and designed the experiments; C.A.G., B.V.-M. and J.G.R.-Q.; contributed reagents/materials/analysis tools; M.M.-M. and L.F.P.-I.; performed the experiments; M.M.-M., C.A.G., J.J.A.-F. and J.G.R.-Q.; analyzed and interpreted the data; M.M.-M.; wrote the manuscript; C.A.G., B.V.-M. and J.G.R.-Q.; revised the manuscript. All authors have read and agreed to the published version of the manuscript.

**Funding:** This research was funded by SENER-CONACYT, grant number 246911.

**Institutional Review Board Statement:** Not applicable.

**Informed Consent Statement:** Not applicable.

**Data Availability Statement:** Not applicable.

**Acknowledgments:** The authors are grateful for the support of: Fondo Sectorial SENER-CONACYT (CEMIE-Bio) Grant No. 246911, Dirección General de Asuntos del Personal Académico of the UNAM, grant UNAM-PAPIIT IA105820, and IBEROMASA Network (719RT0586) of the Ibero-American Program of Science and Technology for Development (CYTED).

**Conflicts of Interest:** The authors declare no conflict of interest.

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
