# Peer review of "Evaluation and Characterization of Timber Residues of Pinus spp. as an Energy Resource for the Production of Solid Biofuels in an Indigenous Community in Mexico"

_forests, doi:10.3390/f12080977_

Round 1
Reviewer 1 Report
The article touches on the important problem of developing a sustainable way to generate energy for the ever expanding global economy. What I miss in the introduction is more emphasis on proposals to deal with this problem based on proposals from other countries/regions. I would suggest adding something on this topic based on the suggested literature below.
- Roman, K.; Barwicki, J.; Hryniewicz, M.; Szadkowska, D.; Szadkowski, J. Production of Electricity and Heat from Biomass Wastes Using a Converted Aircraft Turbine AI-20. Processes 2021, 9, 364. https://doi.org/10.3390/pr9020364
- Demirbas, A.; Competitive liqid biofuels from biomass. Applied Energy 2011, Vol. 88 issue 1, pp.17-28, DOI: https://doi.org/10.1016/j.apenergy.2010.07.016
- Gomez, L.D.; Steele-King C.G.; McQueen-Mason S.J.; Sustainable liquid biofuels from biomass: the writing's on the walls. New Phytologist 2008, Vol. 178, Issue 3, pp. 473-485, DOI: https://doi.org/10.1111/j.1469-8137.2008.02422.x
- Stocker, M.; Biofuels and biomass- to- liquid fuels in biorefinery: Catalytic conversion of lignocellulosic biomass using porous materials, Angewandte Chemie International Edition 2008, Vol. 47, Issues 48, pp. 9200-9211, DOI: https://doi.org/10.1002/anie.200801476
As for the conclusions, I would suggest generalizing them. Lignocellulosic biomass represents one of the greatest potentials for environmentally neutral energy and the example of this community is not an isolated case. Personally I would point out that on the example of the presented community wood waste can be a valuable source of energy allowing to partially or even (in the case presented) completely secure the energy demand of the community. Because biomass, which allows to obtain XYZ energy, is currently not utilized material. Of course, this raw material can also be used in the future to develop fuels with lower emissions, etc.
Author Response
Authors Response to comments of the manuscript: forests, Manuscript ID: forests-1292917
The authors would like to thank each reviewer for his/her constructive comments and suggestions to improve the quality of the manuscript. We have taken into account each comment in the revised version of the manuscript.
A particular response to comments offered by each reviewer is provided next:
Response to Reviewer #1
- The article touches on the important problem of developing a sustainable way to generate energy for the ever expanding global economy. What I miss in the introduction is more emphasis on proposals to deal with this problem based on proposals from other countries/regions. I would suggest adding something on this topic based on the suggested literature below.
Answer: Suggested literature to give greater emphasis to our research proposal was included in the Introduction chapter.
- As for the conclusions, I would suggest generalizing them. Lignocellulosic biomass represents one of the greatest potentials for environmentally neutral energy and the example of this community is not an isolated case. Personally I would point out that on the example of the presented community wood waste can be a valuable source of energy allowing to partially or even (in the case presented) completely secure the energy demand of the community. Because biomass, which allows to obtain XYZ energy, is currently not utilized material. Of course, this raw material can also be used in the future to develop fuels with lower emissions, etc.
Answer: The Conclusions were corrected based on the suggested comments.

Reviewer 2 Report
The article titled ‘Evaluation and characterization of timber residues of Pinus spp., as an energy resource for the production of solid biofuels in an Indigenous Community in Mexico paper; however, some improvement need. Please see the below comments.
- In the abstract, it was only described introduction and results. The abstract should be reflected of the whole paper in brief (Introduction, methods, results, conclusion/recommendation)
- The hypothesis of this article should include and clearly described in the introduction section.
- The results and discussion part should separate. The authors only showed laboratory results and some references reelecting those laboratory tests. In the separate discussion part, it will be better to elaborate production of solid biofuels in an Indigenous Community in Mexico, reflecting some results according to the title.

Author Response
Authors Response to comments of the manuscript: forests, Manuscript ID: forests-1292917
The authors would like to thank each reviewer for his/her constructive comments and suggestions to improve the quality of the manuscript. We have taken into account each comment in the revised version of the manuscript.
A particular response to comments offered by each reviewer is provided next:
Response to Reviewer #2
- In the abstract, it was only described introduction and results. The abstract should be reflected of the whole paper in brief (Introduction, methods, results, conclusion/recommendation)
Answer: We wrote the summary based on the guidelines of the Forests, we also take as an example some publications of the last 5 years:
- Ngangyo-Heya, M.; Foroughbahchk-Pournavab, R.; Carrillo-Parra, A.; Rutiaga-Quiñones, J.G.; Zelinski, V.; Pintor-Ibarra, L.F. Calorific Value and Chemical Composition of Five Semi-Arid Mexican Tree Species. Forests 2016, 7, 1–12, doi:10.3390/f7030058.
- Fern, E.; Carlos, J.; Alexander, D.; Espinel, A. Assessment of Guava ( Psidium Guajava L .) Wood Biomass for Briquettes ’ Production. forests 2018, 9, 1–13, doi:10.3390/f9100613.
- Núñez-Retana, V.D.; Wehenkel, C.; Vega-Nieva, D.J.; García-Quezada, J.; Carrillo-Parra, A. The Bioenergetic Potential of Four Oak Species from Northeastern Mexico. Forests 2019, 10, 869, doi:10.3390/f10100869.
- Reis Portilho, G.; Resende De Castro, V.; Oliveira Carneiro, A.D.C.; Cola Zanuncio, J.; Vinha Zanuncio, A.J.; Gabriella Surdi, P.; Gominho, J.; De Oliveira Araújo, S. Potential of Briquette Produced with Torrefied Agroforestry Biomass to Generate Energy. Forests 2020, 11, 1–10, doi:https://doi.org/10.3390/f11121272.
- Quiñones-reveles, M.A.; Ruiz-García, V.M.; Ramos-vargas, S.; Vargas-Larreta, B.; Masera, O.; Ngangyo-Heya, M.; Carrillo-Parra, A. Assessment of Pellets from Three Forest Species : From Raw Material to End Use. forests 2021, 12, 447, doi:10.3390/f12040447.
- The hypothesis of this article should include and clearly described in the introduction section.
Answer: We have included the hypothesis of our manuscript in lines 109 to 114.
- The results and discussion part should separate. The authors only showed laboratory results and some references reelecting those laboratory tests. In the separate discussion part, it will be better to elaborate production of solid biofuels in an Indigenous Community in Mexico, reflecting some results according to the title.
Answer: We have integrated the Results and the Discussion into a single chapter, and we have based ourselves on some articles recently published in this same journal. As an example are the following papers:
- Ngangyo-Heya, M.; Foroughbahchk-Pournavab, R.; Carrillo-Parra, A.; Rutiaga-Quiñones, J.G.; Zelinski, V.; Pintor-Ibarra, L.F. Calorific Value and Chemical Composition of Five Semi-Arid Mexican Tree Species. Forests 2016, 7, 1–12, doi:10.3390/f7030058.
- Shrestha, S.; Dwivedi, P. Projecting Land Use Changes by Integrating Site Suitability Analysis with Historic Land Use Change Dynamics in the Context of Increasing Demand for Wood Pellets in the Southern United States. forests 2017, 8, doi:10.3390/f8100381.
- Fern, E.; Carlos, J.; Alexander, D.; Espinel, A. Assessment of Guava ( Psidium Guajava L .) Wood Biomass for Briquettes ’ Production. forests 2018, 9, 1–13, doi:10.3390/f9100613.
- Thi, E.; Barrette, J.; Blanchet, P.; Nguyen, Q.N. Optimizing Quality of Wood Pellets Made of Hardwood Processing Residues. forest 2019, 10, 1–19.
